# Non-Opioid Anesthetics Addiction: A Review of Current Situation and Mechanism

**DOI:** 10.3390/brainsci13091259

**Published:** 2023-08-30

**Authors:** Liyun Deng, Lining Wu, Rui Gao, Xiaolin Xu, Chan Chen, Jin Liu

**Affiliations:** 1Department of Anesthesiology, West China Hospital, Sichuan University, Chengdu 610041, China; dengliyundoctor@163.com (L.D.); wlnahmu163@163.com (L.W.); sophiegaorui@foxmail.com (R.G.); drxlxu@sina.cn (X.X.); scujinliu@foxmail.com (J.L.); 2The Research Units of West China (2018RU012)-Chinese Academy of Medical Sciences, West China Hospital, Sichuan University, Chengdu 610041, China

**Keywords:** drug addiction, substance use disorders, non-opioid anesthetics

## Abstract

Drug addiction is one of the major worldwide health problems, which will have serious adverse consequences on human health and significantly burden the social economy and public health. Drug abuse is more common in anesthesiologists than in the general population because of their easier access to controlled substances. Although opioids have been generally considered the most commonly abused drugs among anesthesiologists and nurse anesthetists, the abuse of non-opioid anesthetics has been increasingly severe in recent years. The purpose of this review is to provide an overview of the clinical situation and potential molecular mechanisms of non-opioid anesthetics addiction. This review incorporates the clinical and biomolecular evidence supporting the abuse potential of non-opioid anesthetics and the foreseeable mechanism causing the non-opioid anesthetics addiction phenotypes, promoting a better understanding of its pathogenesis and helping to find effective preventive and curative strategies.

## 1. Introduction

Addiction is a chronically relapsing brain disease, which is related to the complex interactions between biological factors (pharmacological effects of drugs, genetics, epigenetics, developmental attributes, and neurocircuitry) and environmental factors (social and cultural systems, insufficient social connection, stress, trauma, and exposure to alternative reinforcers) [1,2]. The characteristics of addiction are compulsively seeking and taking drugs, loss of control when limiting intake, and negative emotions (such as irritability, anxiety, and dysphoria) when drugs are unavailable [3,4]. Although opioids have been reported to be the most abused anesthesia medication, the trend of non-opioid anesthetics abuse has become increasingly severe in recent years [5,6]. 

The non-opioid anesthetics have gradually become a significant cause of addiction, death, and career termination [6]. Recently, a retrospective study among anesthesiologists and nurse anesthetists in Australia and New Zealand shows that the pattern of drug abuse has changed compared with that before. Specifically, propofol has replaced opioids as the most frequently abused substance. Moreover, other non-opioid anesthetics include ketamine, benzodiazepines, and inhalational anesthetics [5,7,8]. Non-opioid anesthetics have gradually become an essential source of drug abuse. However, their incidence, related adverse events, and mortality have been underestimated. Furthermore, a comprehensive understanding of the neurobiological mechanisms of establishment and maintenance of non-opioid anesthetics addiction is still lacking. 

The purpose of this review is to provide an overview of non-opioid anesthetic drug addiction. We started with a brief overview of the clinical situation and then introduced the molecular mechanism of non-opioid anesthetic drug addiction (including propofol, ketamine, benzodiazepines, and inhaled anesthetics), providing insights into the potential novel therapeutic target of non-opioid anesthetic drug addiction.

## 2. Methods

The manuscript includes research articles gathered from PubMed that expound on the clinical situation and mechanism of non-opioid anesthetics addiction. The following keywords were used: “anesthesiology”, “drug addiction”, “substance use disorders”, “propofol addiction”, “ketamine addiction”, “benzodiazepines addiction”, and “inhalational anesthetics addiction”. Only articles written in English were selected for study.

## 3. The Current Situation of Non-Opioid Anesthetics Addiction

Among the non-opioid anesthetics abused by anesthesiologists, propofol, ketamine, benzodiazepines, and inhaled anesthetics topped the list. We review the current prevalence of these non-opioid anesthetic medications used by anesthesia personnel.

Propofol is one of the most common intravenous anesthetics. Due to its rapid onset, stable efficacy, and short duration, it has been widely used in clinical practice [9,10]. It can be used for induction and maintenance of general anesthesia, long-term sedation, monitored anesthesia care (MAC), and conscious sedation in the intensive care unit (ICU) [11]. However, the abuse potential of propofol has been underestimated, and it has not been a controlled substance anywhere in the world except in Korea so far [12]. It is reported that the number of propofol abuse increased five times from 1995 to 2005 [13]. In a retrospective study, propofol has been reported to become the most frequent abuse of anesthesia drugs among Australian and New Zealand anesthetists, accounting for 41% of drug abuse cases from 2004 to 2013 [7]. Notably, the safe margin between the euphoric dose and the lethal dose caused by propofol is relatively narrow, and many abusers die of respiratory depression caused by excessive propofol, and the mortality rate can be as high as 45% [14].

Ketamine is a noncompetitive NMDAR antagonist, first synthesized by American chemist Calvin Lee Stevens in the Parke-Davis laboratory in 1962 and approved by the US Food and Drug Administration (FDA) in 1970 [15]. Ketamine is often used for anesthesia in children, superficial surgery, and burn operations and can also be used as an adjunct to regional block anesthesia [16]. Although ketamine can inhibit the thalamic-neocortical, it can activate the limbic system. It provides a unique state of anesthesia in which consciousness and physical sensation are temporarily separated (the so-called “dissociative anesthesia”), which is the basis of ketamine abuse [17]. The patient feels relaxed and euphoric after obtaining ketamine, but their consciousness remains clear. Before 1990, ketamine accounted for 4% of anesthetic drug addiction [18]. From the 1990s to the beginning of the 21st century, ketamine accounted for approximately 2% of the drugs abused among American anesthesiologists, and it was placed into Schedule III of Schedules of Controlled Substances by the US Drug Enforcement Administration [19,20]. Long-term ketamine use can cause permanent damage to the brain, as well as negative effects on the urinary, cardiovascular, and digestive systems [21,22,23,24].

Benzodiazepines (BDZs) have sedative, hypnotic, and anti-anxiety effects, which are usually used as pre-anesthetic drugs, adjuvants for local and intraspinal anesthesia, induction and maintenance of general anesthesia in the maintenance practice [25,26]. BDZs are among the most commonly abused non-opioid anesthetic drugs, accounting for 14% of the drugs abused among anesthesiologists in the United States [19]. It is often accompanied by the misuse of other substances, making its treatment very complex. Midazolam is the most commonly abused BDZs [5]. Long-term use of BDZs can cause cognitive and motor disorders and severe withdrawal symptoms (including anxiety, insomnia, muscle spasms, nervousness, and hypersensitivity) [27,28].

Inhalational anesthetics have been reported to account for 2% of drug abuse among anesthesiologists in the United States and 5% in Australia and New Zealand [7,29]. Though inhalational anesthetics were not generally regarded as addictive drugs in the past, recently, more and more reports have shown that they also have abuse potential [30,31]. In a survey of inhalational anesthetic abuse in anesthesia training programs, nitrous oxide (N_2_O), isoflurane, sevoflurane, halothane, and desflurane accounts for 47%, 24%, 19%, 19%, and 9.5%, respectively. Since its discovery and first use in the 1840s, N_2_O has been widely used for anesthesia in dental and cesarean operations. Nowadays, N_2_O use disorder has been well-documented [32,33]. One of the most concerning aspects of inhalational anesthetics addiction is the high mortality, which is similar to propofol abuse [34]. Acute N_2_O can lead to severe hypoxia symptoms [34,35]. Long-term exposure to N_2_O can lead to many psychiatric symptoms, such as delusions, delirium, and depression. Apart from psychiatric symptoms, it also causes B_12_ deficiency, leading to peripheral neuropathy and megaloblastic anemia [34,35,36,37,38,39,40,41,42].

## 4. Molecular Mechanism of Non-Opioid Anesthetic Drug Addiction 

Drug addiction is characterized by the damage and disorder of multiple neural networks and circuits in the brain, which includes the reward system, the anti-reward/stress system, and the central immune system [43]. Specifically, the reward system has a crucial role in driving drug-seeking behavior. The reward system is a regional network composed of the limbic system, prefrontal cortex (PFC), striatum, and midbrain [44], and the dopaminergic (DAergic) delivery from the ventral tegmental area (VTA) to the nucleus accumbens (NAc) is the basis of the reward system [2]. Moreover, DAergic, glutamatergic (GLUergic), and GABAergic delivery in the PFC, hippocampus, amygdala, and globus pallidus are also closely related to drug-seeking behavior, withdrawal symptoms, and negative emotional responses [3,45,46] (Figure 1A). Dopamine (DA) is located in the center of drug reward signal transduction in the brain and a common feature of different abused drugs is that they eventually cause the increase of DA in NAc through different pharmacological effects [44]. With the continuous research and deep understanding of reward circuits, the research on neurotransmitters and neuromodulators of drug addiction has also been deepened. Recent studies have found that, in addition to DA, other neurotransmitters and neuromodulators such as glutamate, gamma-aminobutyric acid (GABA), opiate peptides, serotonin, acetylcholine, and endogenous cannabinoids also play a vital role in drug addiction and relapse [47,48,49,50]. Although they are not widely studied comparable to DA, their role in addiction should not be underestimated. Changes in synaptic plasticity caused by drug-induced DA increase is another common feature of addictive drugs [45] (Figure 1B). Increased DA release can trigger various forms of synaptic plasticity changes by altering the expression of genes closely related to synaptic plasticity, such as ΔFosB, NFκB, etc., or by RNA editing, leading to the strengthening or weakening of connections between different nuclei in the reward circuit [51]. The changes in synaptic plasticity mainly include the plasticity of morphological structure and function. The morphological plasticity mainly includes the changes in the synaptic number, synaptic interface structure, and dendrites. Functional plasticity mainly refers to long-term potentiation (LTP) and long-term depression (LTD), which underlie learning and memory [45,51].

Although the additive drugs may converge on a common addition pathway, the primary target receptors or proteins of different drugs are not the same. Thus, exploring the specific cellular and molecular mechanisms involved in non-opioid anesthetics addiction is necessary to identify effective therapy.

### 4.1. Propofol

In animal experiments, the addictive potential of propofol has been confirmed by two classical behavioral methods for assessing drug abuse (namely self-administration (SA) and conditioned place preference (CPP)) [52,53,54,55,56]. Similar to other addictive drugs, repeated propofol exposure can activate the reward system [57]. Many studies have explored the possible mechanism of propofol addiction at the neural circuit, molecular, and cellular levels (Table 1).

#### 4.1.1. Propofol Addiction and DAergic System

The mesolimbic DA system is also recognized as the most important brain region related to addiction. DA is a neurotransmitter that can bring pleasure to people and is thought to be at the center of the reward circuit, which is often considered the “culprit” of addiction [43,66,67,68]. Drugs with addictive potential can increase the release of DA by direct or indirect effects according to their distinct pharmacological effects on different molecular targets [2,51]. DA signaling is mainly mediated by DA receptors (DARs) and several downstream signaling molecules. DARs belong to the G protein-coupled receptor (GPCP) superfamily and DARs include five members: Dopamine receptor 1 (DAR1), Dopamine receptor 2 (DAR2), Dopamine receptor 3 (DAR3), Dopamine receptor 4 (DAR4), and Dopamine receptor 5 (DAR5) [69,70]. In the striatum, DA mainly interacts with DARs on medium spiny neurons (MSN). The MSN in the striatum can be divided into two subgroups: substantia nigra neurons in the direct striatal pathway and pallidum neurons in the striatal indirect pathway, and there are huge differences in the expression of functional and signaling molecules between these two subgroups [71]. The direct and indirect pathways in the striatum play distinct roles in regulating the reward and motivation of addictive drugs. It is generally believed that the direct pathway is the basis of reward, while the indirect pathway is related to punishment [72]. In the direct pathway, DA binds to DAR1, in a stimulatory manner, which can mediate reward through increasing cyclic adenosine monophosphate (cAMP) and intracellular calcium signal transduction; In the indirect pathway, DA binds to DAR2 in an inhibitory manner, specifically reducing the cAMP and intracellular calcium signal transduction as well as opposing aversive responses [2,73]. In addition, DA also interacts with DAR3 and DAR4 that co-locate with DAR1, exerting significant influence on DAR1 signaling and neural plasticity. The downstream signal molecules of DARs mainly include protein kinase A (PKA), protein kinase C (PKC), extracellular signal regulated kinase (ERK), calmodulin dependent protein kinase II (CaMKII), etc. These protein kinases are generally believed to affect the transmission function of synapses by changing the properties or density of ion channels, and regulate the structure of synapses both at the gene and protein levels [74,75]. 

Based on a microdialysis probe, Pain and colleagues found propofol could also alter the level of DA in the NAc [76]. Similarly, another study found a significant increase in the expression of dopamine in the thalamus [77]. Repeated propofol abuse also upregulated DAR1 and its downstream signaling molecules DeltaFosB and p-ERK in the NAc, while the antagonist of DAR1 (SCH23390) and ERK (U0126) attenuated the self-administration of propofol [54,58,59]. These results suggested that the DAergic system in the NAc is involved in the maintenance of propofol self-administration and its rewarding effects (Figure 2A).

#### 4.1.2. Propofol Addiction and Stress/Anti-Reward System

While the reward system is related to positive reinforcing, the anti-reward system is related to negative reinforcing in the central nervous system (CNS). The changes in the reward system and the anti-reward system eventually led to the formation and recurrence of addictive behaviors [78,79]. The extended amygdala located at the basal forebrain which is critical for the anti-reward system has been implicated in the experience of negative emotions associated with drug withdrawal. The extended amygdala is composed of the bed nucleus of stria terminals (BNST), the NAc, and the central amygdaloid nucleus (CeA) [78,80]. The neurotransmitters involved in the anti-reward system include corticotropin-releasing factor (CRF), norepinephrine, dynorphin, and neuropeptide Y, among which the CRF system is one of the best-studied neurobiochemical basis of the anti-reward system [80,81]. 

CRF plays its role through CRF receptor-1 (CRF1R) and CRF receptor-2 (CRF2R). As Dong indicated in a 2010 study, the antagonist of CRF1R significantly reduced the self-administration behavior of propofol [53]. In addition, direct injection of glucocorticoid receptor (GR) agonists into the NAc promoted propofol addiction in rats, and the self-administration of propofol could also be blocked by GR antagonists [60,61], suggesting that the CRF system plays an essential role in propofol drug-seeking behavior (Figure 2B).

#### 4.1.3. Propofol Addiction and Nitrergic System

Nitric oxide (NO) is a bioactive free radical produced by most cells. The endogenous pathway of NO synthesis is mainly produced by the two-step oxidation of L-arginine catalyzed by nitric oxide synthase (NOS) [82,83,84]. In previous studies, NO was generally considered to be a toxic intermediate that was involved in a variety of disease processes. However, in recent years, there has been increasing attention to its physiological functions as a key messenger molecule in the central nervous system (CNS). For example, it regulates the release and uptake of neurotransmitters such as DA and glutamate and participates in learning and memory processes [83,84].

In the past years, an increasing number of studies have shown the significance of NO in the process of drug reward. In the animal model of propofol addiction, peripheral administration of NOS inhibitors significantly inhibited the conditioned spontaneous activity of rats [55,62]. At the same time, it also reverses propofol-induced CPP, suggesting that the central nitrergic system may be one of the causes of propofol-induced CPP [63] (Figure 2C).

In addition to the aforementioned mechanisms, recent studies have also found that N-methyl-D-aspartate receptor (NMDAR) and adenosine A2A receptor (A2AR), which were previously thought to be related to the dependence of classic addictive drugs such as opioids and cocaine, can also regulate the addictive behavior of propofol [52,65].

### 4.2. Ketamine

Ketamine has a strong addiction potential via inducing SA and CPP [85,86,87]. However, there is no systematic review of the specific mechanisms of ketamine addiction. As below, we describe the possible mechanisms of ketamine addiction (Table 2).

#### 4.2.1. Ketamine Addiction and GLUergic System

Growing evidence has identified the glutamate system as a crucial role in mediating drug addiction [105,106,107,108,109]. Glutamate is a major excitatory neurotransmitter in the CNS. Its receptors can be divided into two categories: ionic receptors, including NMDAR, kainate receptor (KAR), α-amino-3-hydroxy-5-methyl-4-isox-azolepropionic acid receptor (AMPAR); the other is metabotropic glutamate receptors (mGluRs). In these two types of receptors, NMDAR, AMPAR, and mGluRs are closely related to drug addiction [43,110,111].

Ketamine, as a blocker of NMDAR, may lead to addiction mainly by inhibiting the activity of NMDAR. NMDAR is a tetrameric ion channel, which is composed of two obligatory GluN1 subunits and two GluN2 or GluN3 subunits, of which GluN2 have four (GluN2A-GluN2D) and GluN3 have two (GluN3A GluN3B) subtypes, respectively [112,113]. As a non-competitive antagonist of NMDAR, ketamine can block NMDAR on GABAergic neurons, causing the disinhibition of DAergic neurons and the activation of the reward system [88,114]. In addition, repeated administration of ketamine significantly increased the expression of the GluN1 subunit gene and increased the polymorphism of the GluN2B gene [115,116]. Moreover, ketamine-induced hyperlocomotion and behavioral sensitization would be blocked by the knockout of the GluN2D gene, indicating that GluN2D plays a regulatory role in ketamine addiction [89,91].

#### 4.2.2. Ketamine Addiction and DAergic System

The mesolimbic DA system has been implicated in ketamine abuse and potentiation, with increased DA release following repeated ketamine use [117,118]. Continuous stimulation of DAR2 by excessive DA can reduce the number and sensitivity of DAR2. The decrease in DAR2 level gradually reduces the ability to receive rewards, thus increasing the demand for ketamine, further leading to the low reaction of DAR2, and forming a vicious circle [92]. Moreover, pretreatment with a DAR2 antagonist significantly blocked the addictive behavior of the mice, indicating that DAR2 plays a regulatory role in ketamine addiction [93]. Furthermore, inhibition of ERK and CREB phosphorylation, downstream signal molecules of DARs, can inhibit the acquisition of ketamine-induced conditioned place preference [95,96].

#### 4.2.3. Ketamine Addiction and Changes in Brain Structure and Functional Brain Network Integrity

Non-invasive neuroimaging has contributed important insights into identifying the differences in brain structure and function between substance use disorders (SUD) patients and healthy individuals. Using structural magnetic resonance imaging (sMRI), researchers found that the brain structure of ketamine addicts had distinct changes, including significant atrophy of the frontal, parietal, or occipital cortex [98,119].

Recent years have witnessed the explosive growth of functional magnetic resonance imaging (fMRI) in studying brain functional connectivity, especially resting state functional connectivity (rsFC) [120,121,122]. The rsFC can provide a valuable tool for exploring large-scale brain networks and their interactions and identifying the neural circuit disorders behind SUD [121,123]. Using fMRI, we found that ketamine could significantly increase neuronal activity, especially in the PFC, anterior cingulate cortex, precuneus, and precentral frontal gyrus [99,100,101]. Chronic ketamine use can also alter rsFC, especially in the thalamus, basal ganglia, and higher cortex [97]. Ketamine abusers showed decreased thalamocortical connectivity. Specifically, compared with control subjects, ketamine users had significantly reduced connectivity between thalamic nuclear and PFC, motor cortex, and posterior parietal cortex [97]. In another study, ketamine altered rsFC in the striatum, significantly increasing connectivity between the caudate nucleus and the dorsal anterior cingulate cortex, between the pallidum and bilateral cerebellum, and between the putamen and the left orbitofrontal cortex [102]. The increased connection between the caudate nucleus and dorsal anterior cingulate cortex may be related to drug craving, while the connection between the putamen and OFC may increase impulsive drug-seeking behavior [124,125,126,127]. In conclusion, these findings suggest that chronic ketamine use can change the structure and functional connectivity of brain regions related to reward processing.

In addition to the above mechanisms, some studies have shown that ketamine can cause changes in synaptic plasticity in the NAc and hippocampus [87,96]. However, a recent study published in *Nature* suggested that ketamine has rewarding properties, but it could not induce synaptic plasticity similar to other typical addictive drugs, so ketamine has low addiction liability [88]. However, the true picture of ketamine addiction is probably more complex and does not arise through a single pathway. For example, glial cells play a key role in forming neuronal synaptic activity, and ketamine may have an effect on the glial [128]. What is more, long-term use of ketamine use may also contribute to an increased risk of addiction through its effects on other brain regions and cell types. Therefore, further research is needed to determine the risk of addiction to long-term ketamine use.

### 4.3. Benzodiazepines

The addictive drugs can be divided into three groups through their mechanism of increasing DA in the midbrain margin:(1) some drugs can reduce the inhibitory afferent of gamma GABAergic neurons to DAergic neurons, which is defined as disinhibition (opioids, γ-hydroxybutyric acid, and cannabinoids). (2) The second group can depolarize DAergic neurons directly by activating the α4β2 acetylcholine receptor (nicotine). (3) The third group is those drugs that can target DA transporters and interfere with DA reuptake (cocaine, amphetamines, and ecstasy) [66,129]. Similar to opioids, γ-hydroxybutyric acid, and cannabinoids, BDZs also increase DA levels through disinhibition, but their molecular target is different. BDZs is an allosteric modulator of GABA type A receptors (GABA_A_Rs). According to their efficacy, BDZs can be divided into three groups: positive allosteric modulators, negative allosteric modulators, and antagonists [130]. Most of the clinically classical BDZs, including midazolam, are positive allosteric modulators of GABA_A_Rs, which can enhance the function of GABA_A_Rs through conformational changes and increase the affinity for GABA [131]. GABA_A_Rs are usually composed of at least 19 subunits (α 1-6, β 1-3, γ 1-3, δ, ε, θ, π, ρ 1-3), which is a ligand-gated chloride channel [131]. Receptors containing α1, α2, α3, or α5 subunits can be modulated by classical BDZs whereas receptors containing α4 and α6 subunits isoforms are insensitive to BDZs [130,132,133,134]. BDZs bind to a pocket formed by the α and the γ subunits [131,135]. Since most of the pharmacological effects of BDZs are mediated through binding to GABA_A_Rs, changes in the number, structure, and function of GABAARs may play a crucial role in the development and progression of BDZ addiction (Table 3).

#### 4.3.1. BDZs Addiction and α1-GABA_A_Rs

Recent studies have shown that α1-GABA_A_Rs are closely related to the addictive properties of BDZs [137,144]. VTA is an important brain region in the DA pathway, mainly composed of DAergic neurons (about 70%), GABAergic neurons (about 20%), and GLUergic neurons (about 10%) [145]. Normally, the activity of DAergic neurons is inhibited by GABAergic interneurons [145]. Both DAergic neurons and GABAergic interneurons express GABA_A_Rs, but the α1 subunit is specifically expressed on GABAergic neurons while α2, α3, and α4 subunits are expressed on DAergic neurons [132,146]. Longtime exposure to BDZs can lead to the combination of BDZs with GABA_A_Rs on these two kinds of neurons and enhance inhibitory GABA-induced currents, but the effect on GABAergic interneurons was greater than that on DAergic neurons. The increased inhibitory currents of interneurons led to the reduction of GABA release, and thus its inhibitory effect on DAergic neurons were relieved, ultimately leading to more DA release [132,136] (Figure 3).

#### 4.3.2. BDZ Addiction and Other Subtypes of GABA_A_Rs

Although α1-GABA_A_Rs are involved in the addictive process of BDZs, it was found that the animals could still maintain the drug administration behavior when animals were treated with L-838, 417, the antagonist of α1-GABA_A_Rs, suggesting that other GABA_A_Rs subtypes may also be involved in the process of BDZs abuse [141]. Using H-R mutant mice, researchers found that the reward effects of diazepam were reduced in α1 subunit mutation (H101R) mice, but the reward effects were abolished in α2 subunit mutation (H101R) mice. These results demonstrated that both α1-GABA_A_Rs and α2-GABA_A_Rs may be involved in BDZs addiction, and the role of α2-GABA_A_Rs may be more dominant [139]. Furthermore, the recombinant adeno-associated virus was used for the knockdown of α2-GABA_A_Rs, and it was shown that the preference for midazolam depends on the positive modulation of α2-GABAARs in the NAc [139]. Similarly, another study using an intracranial self-stimulation (ICCS) test and point mutation mice also suggested that α2-GABA_A_Rs and α3-GABA_A_Rs were the key mediators of BDZs reward effects [138]. In addition, α5-GABAARs receptors in the dentate gyrus appear to be specific targets associated with BDZ tolerance through α5 subunit mutation mice [143]. Furthermore, it has been confirmed in primates that the activity of α2-GABA_A_Rs, α3-GABA_A_Rs, and α5-GABA_A_Rs is related to the addictive potential of BDZs [141]. In conclusion, apart from α1-GABA_A_Rs, other subtypes of GABA_A_Rs are also crucial in the addiction of BDZs, and these findings provide a solid foundation for the development of new drugs for treating BDZs abuse.

### 4.4. Inhalational Anesthetics

Although there are increasing studies emphasizing the addictive properties of inhalational anesthetics, research on its specific mechanisms is lacking [147,148]. Studies have found that almost all addictive drugs can cause an increase in dopamine release in the NAc, and N_2_O is no exception. Through rat microdialysis studies, it was found that the DA levels in NAc of rats exposed to N_2_O were significantly increased, accompanied by the phosphorylation of ERK [149]. The antagonist of DAR1 was found to inhibit N_2_O-induced CPP and enhance p-ERK, indicating that DAR1 plays an important role in N_2_O reward [150]. Together, these findings suggest that the reward effects of N_2_O are closely linked to the dopaminergic system.

## 5. Conclusions and Future Perspectives

Increasing evidence have indicated that the incidence of abuse of non-opioid anesthetic drugs (including propofol, ketamine, BDZs, and inhalational anesthetics) is increasing, which needs more attention from researchers and exposed patients. Effective regulatory and therapeutic methods are needed for the identification, monitoring, and treatment of individuals with non-opioid anesthetic drug abuse. Understanding the current situation and mechanism of non-opioid anesthetics addiction is helpful to improve a better understanding of it and can provide certain ideas for the prevention and treatment of non-opioid anesthetics. Growing studies have confirmed the abuse potential for propofol, ketamine, BDZs, and inhalational anesthetics using CPP and SA models. Similar to other classic addictive, propofol, ketamine, BDZs, and inhalational anesthetics could also influence the reward system. However, the focus on the clinical situation and the potential mechanisms of non-opioid anesthetics addiction appears to be in its preliminary stages.

In the future, more clinical studies are needed to update the incidence of non-opioid anesthetics addiction and determine the population at risk of non-opioid anesthetics addiction abuse. In addition, more advanced imaging techniques, tracing techniques, optogenetic tools, chemogenetic tools, and electrophysiological techniques should also be used to analyze the complex neural network behind the specific reward circuitry of non-opioid anesthetics to further clarify the specific mechanism of the non-opioid anesthetics addiction.

## Figures and Tables

**Figure 1 brainsci-13-01259-f001:**
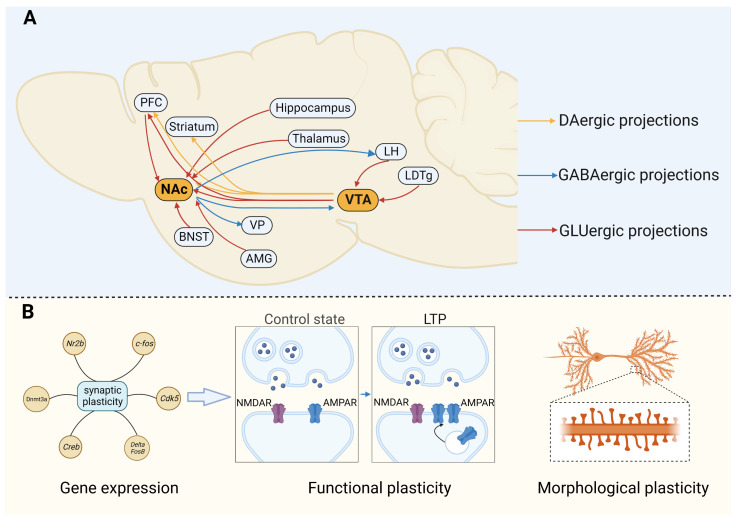
Common characteristics of addictive drugs. (**A**) Schematic representation of the brain reward system. The DAergic delivery from the VTA to the NAc is located at the center of the reward system. VTA DAergic neurons can also project to the PFC and striatum and receive excitatory glutamate inputs from the LH and the LDTg. Meanwhile, GABAergic medium spiny neurons in the NAc receive GLUergic inputs from the PFC, BNST, hippocampus, thalamus, and amygdala, and send GABAergic projections to the LH, VTA, and VP. (**B**) Schematic representation of the changes of synaptic plasticity. Persistent drug exposure can cause changes in synaptic plasticity, which initially causes changes in the levels of fosB, DeltaFosB, NFκB, CdK5, DNMT3A, and MEF2 genes, and then repeated drug exposure can cause changes in the morphological structure and function of the synapse. Functional plasticity mainly includes LTP of synaptic transmission and morphological plasticity is mainly reflected in the change of dendritic spine density. AMG, amygdala; AMPAR, alpha-amino-3-hydroxy-5-methyl-4-isoxazolepropionic acid receptor; BNST, bed nucleus of stria terminalis; LDTg, laterodorsal tegmental nucleus; LH, lateral hypothalamus; LTP, long-term potentiation; NAc, nucleus accumbens; NMDAR, N-methyl-D-aspartate receptor; PFC, prefrontal cortex; VP, ventral pallidum; VTA, ventral tegmental area.

**Figure 2 brainsci-13-01259-f002:**
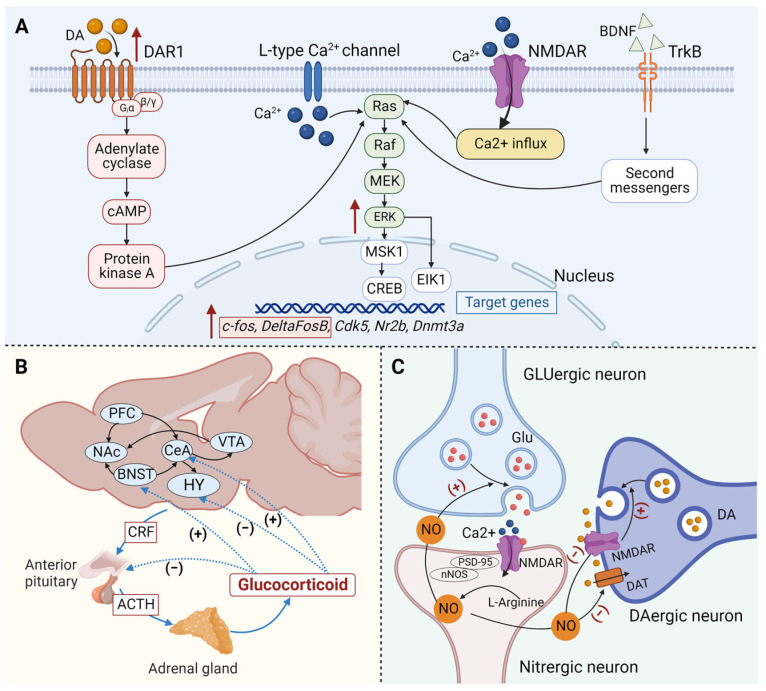
Potential molecular mechanisms underlying propofol addiction. (**A**) Dopamine signal transduction pathway in the brain of propofol-addicted rats. Propofol can promote the release of DA and activate DAR1, leading to the activation of protein kinase A and L-type calcium channels, thus activating the Raf. At the same time, propofol can activate NMDAR, leading to Ca^2+^ influx and Raf activation. After the activation of Raf, it causes the phosphorylation of MEK and ERK, followed by the activation of downstream MSK1 and EIK1, and finally causes the transcription of c-fos, DeltaFosB, and other genes, being a potential mechanism for inducing propofol addiction. (**B**) Schematic representation of the dysregulation of the anti-reward system affecting propofol addiction. The center of the anti-reward system is located in the extended amygdala region of the basal forebrain, including the BNST, NAc, CeA, etc. The CRF system is the neurobiochemical basis of the anti-reward system. The addictive behavior of propofol can be significantly regulated by CRF1R and GR. (**C**) The nitrergic system affects propofol addiction. Glu released by GLUergic neurons promotes the production of NO through NMDAR on the nitrergic neurons. A low level of NO can enhance the release of DA by enhancing the release of Glu. In addition, NO can also reduce the reuptake of dopamine by inhibiting DAT, eventually leading to the increase of extracellular DA. However, excessive NO will inhibit the NMDAR and block the release of DA. ACTH, Adrenocorticotrophin hormone; BDNF, brain-derived neurotrophic factor; BNST, bed nucleus of stria terminalis; cAMP, cyclic adenosine monophosphate; CeA, central amygdaloid nucleus; CRF, corticotropin-releasing factor; CRF1R, CRF receptor-1; DA, dopamine; DAergic, dopaminergic; DAR1, Dopamine receptor 1; DAT, dopamine transporter; Glu, glutamate; GLUergic, glutamatergic; GR, glucocorticoid receptor; HY, hypothalamus; NAc, nucleus accumbens; NMDAR, N-methyl-D-aspartate receptor; nNOS, Neuronal nitric oxide synthase; NO, nitric oxide; PFC, prefrontal cortex; VTA, ventral tegmental area.

**Figure 3 brainsci-13-01259-f003:**
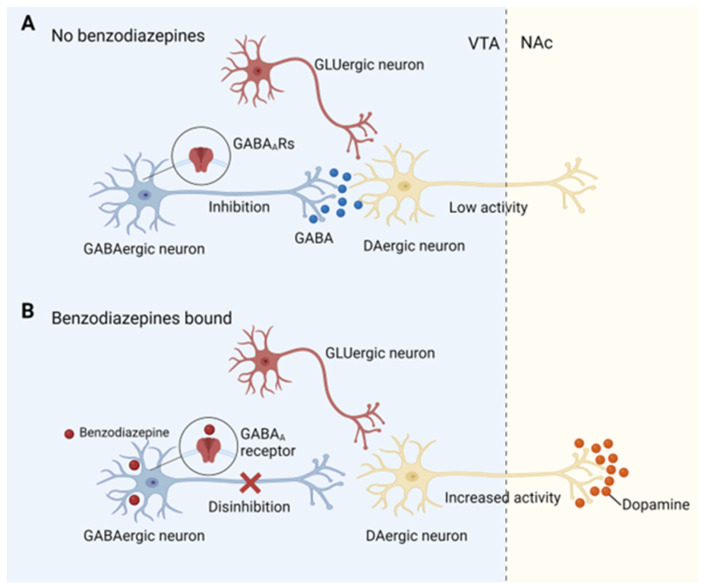
Mechanisms of benzodiazepine addiction. (**A**) Normally, DAergic neurons in the VTA receive inhibitory inputs from inhibitory GABAergic interneurons. As a result, the activity of DAergic neurons is inhibited, and eventually, the release of DA in the NAc is reduced. (**B**) When benzodiazepines bind to α1-GABA_A_Rs receptors on gabaergic neurons, the inhibitory effect of GABAergic neurons on DAergic neurons is relieved (disinhibition), leading to the activation of DAergic neurons and an increase in DA release in the NAc. DA, dopamine; DAergic, dopaminergic; GABA, gamma-aminobutyric acid; GABA_A_Rs, GABA type A receptors; GLUergic, glutamatergic; NAc, nucleus accumbens; VTA, ventral tegmental area.

**Table 1 brainsci-13-01259-t001:** Summary of experiments addressing mechanisms of propofol addiction.

Target	Brain Region	Paradigm	Manipulation	Results	Species	Ref.
ERK	NAc	SA	MEK inhibitor	Impaired propofol-maintained SA	Rat	[54]
ERK	Nac	SA	MEK inhibitor	Decreased SA	Rat	[58]
DeltaFosB	NAc	/	WB, PCR	Increased DeltaFosB expression	Rat	[59]
CRF1R	Systemic	SA	CRF1R antagonist	Inhibited acquisition	Rat	[53]
GR	Systemic	SA	GR agonist or antagonist	Inhibited reward-enhancing effect	Rat	[60]
GR	NAc	SA	GR agonist	Increased SA	Rat	[61]
CNS	Systemic	LA	NO inhibitor	Inhibited LA	Rat	[62]
NO	Systemic	CPP	NOS inhibitor	Abolished CPP	Rat	[55]
NO	Systemic	LA	NOS inhibitor	Inhibited LA	Rat	[63]
GABAAR	Systemic	Nose-poke discrimination	GABAARs agonist	discriminative stimulus effects	Rat	[64]
NMDAR	Systemic	SA	NDMAR antagonist	Inhibited acquisition	Rat	[65]
A2AR	NAc	SA	A2AR agonist or antagonist	Agonist inhibited and antagonist promoted SA	Rat	[52]
NA system	LC	CPP	Chemogenetic inhibition	Abolished CPP	Rat	[56]

A2AR, adenosine A2A; CNS, central nitrergic system; CPP, conditioned place preference; CRF1R, corticotropin-releasing factor-1 receptor; ERK, extracellular regulated protein kinases; GABAAR, gamma-aminobutyric acid type A receptor; GR, glucocorticoid receptor; LA, Locomotor activity; LC, locus coeruleus; NAc, nucleus accumbens; NA, Noradrenergic; NMDA, N-methyl-D-aspartate; NMDAR, NMDA receptors; NO, nitric oxide; NOS, nitric oxide synthase; SA, self- administration.

**Table 2 brainsci-13-01259-t002:** Summary of experiments addressing mechanisms of ketamine addiction.

Target	Brain Region	Paradigm	Manipulation	Results	Species	Ref.
NMDA	VTA	SA, CPP	NR1 knockout	Confined addiction liability	Mouse	[88]
GluN2D	Systemic	/	GluN2D knockout	No changes in sEPSC frequency	Mouse	[89]
GluA2/3	Systemic	/	Rhy	Suppressed glua2/3 and glun1 expression	Rat	[90]
GluN2D	Systemic	LA	GluN2D knockout	Inhibited locomotor sensitization	Mouse	[91]
DAR2, DAT	PFC	/	/	Decreased in DAR2 and DAT expression	Monkey	[92]
5HT2, DAR2	Systemic	LA	5HT2, DAR2 antagonist	Inhibited locomotor sensitization	Mouse	[93]
DA	NAc	/	Pentobarbital	Inhibited ketamine-induced dopamine	Rat	[94]
ERK, CREB	Systemic	CPP	l-THP	Abolished CPP	Rat	[95]
CREB, Nurr1, BDNF	Systemic	CPP	Rhy	Abolished CPP	Rat	[96]
TH	TH	/	rsfMRI	Decreased TH Connectivity	Human	[97]
Cortex	Cortex	/	MRI	Cortical atrophy	Human	[98]
GLUergic system	mFPC	/	MRI	Increased GLUergic activity	Human	[99]
ACC, PC	ACC, PC	/	fMRI	Increased activation in the ACC and PC	Human	[100]
PFC	PFC	/	fMRI	Alterations in the FC of PFC	Human	[101]
Striatal	Striatal	/	fMRI	Altered striatal connectivity	Human	[102]
miR-331-5p	HIP	CPP	Rhy	Abolished CPP	Rat	[103]
αCaMKII	Reward-Related	SA	/	αCaMKII autophosphorylation	Rat	[85]
Spine	NAc	CPP	/	Increased in spine density	Rat	[87]
GSK-3β	CPU, NAc, and VTA	SA	GSK-3β inhibitor	Inhibited reward-enhancing effect	Rat	[104]

ACC, anterior cingulate cortex; BDNF, brain-derived neurotrophic factor; CPP, conditioned place preference; CPU, caudate putamen; CREB, cyclic AMP response element binding protein; DA, dopamine; DAR2, Dopamine receptor 2; DAT, dopamine transporter; ERK, extracellular signal-regulated kinase; FC, functional connectivity; fMRI, functional magnetic resonance imaging; GSK-3β, glycogen synthase kinase-3; HIP, hippocampus; mFPC, medial prefrontal cortex; LA, locomotor activity; l-THP, Levo-tetrahydropalmatine; MRI, magnetic resonance imaging; NAc, nucleus accumbens; NMDA, N-methyl-D-aspartate; Nurr1, nuclear receptor-related-1; PC, precuneus; PFC, prefrontal cortex; PFC, prefrontal cortices; Rhy, Rhynchophylline; rsfMRI, resting-state functional magnetic resonance imaging; SA, self-administration; sEPSC, spontaneous excitatory postsynaptic currents; TH, thalamus; VTA, ventral tegmental area.

**Table 3 brainsci-13-01259-t003:** Summary of experiments addressing mechanisms of BDZs addiction.

Subtypes of GABAARs	Brain Region	Paradigm	Manipulation	Results	Species	Ref.
α1	VTA	SA	Electrophysiology	Disinhibition	Mouse	[136]
α1	Systemic	OCB	GABAARs agonist	Decreased response rates	Monkey	[137]
α2 and α3	NAc	ICSS	α2 and α3-point-mutant mice	Abolished reward-enhancing effect	Mouse	[138]
α2	NAc	ICSS, TBCD	α2-point-mutant mice	Inhibited reward-enhancing effect	Mouse	[139]
α3	Systemic	SA	GABAARs agonist	Reinforced reward-enhancing effect	Monkey	[140]
α2, α3, and α5	Systemic	SA	GABAARs agonist	Reinforced reward-enhancing effect	Monkey	[141]
α2 and α3	Systemic	SA	GABAARs agonist	Reinforced reward-enhancing effect	Baboons	[142]
α5	HIP	LA	α5-point-mutant mice	Reinforced tolerance	Mouse	[143]

BDZs, benzodiazepines; HIP, hippocampus; ICSS, intracranial self-stimulation; LA, Locomotor activity; NAc, nucleus accumbens; OCB, Operant conditioning behavior; SA, self-administration; TBCD, two-bottle choice drinking; VTA, ventral tegmental area.

## Data Availability

The data that support the findings of this study are available from the corresponding author upon reasonable request.

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
