# Peer review of "Non-Opioid Anesthetics Addiction: A Review of Current Situation and Mechanism"

_brainsci, 2023, doi:10.3390/brainsci13091259_

Round 1

Reviewer 1 Report

brainsci-2545944

The authors describe non-opioid drug addiction in anesthesia care providers. Although it may be of little interest to the general public, it is a topic of great interest to health care providers exposed to these drugs. The authors seemingly have a good overview of the landscape of non-opioid anesthetic addiction. However, there seems to be a logical contradiction here and there. This paper generally based on statistics from more than 10 years ago. If the authors use “current“ in the title, it should be supported by more recent (less than 5 years) data.

Comments;

1. p1, line11 “Drug abuse is more severe in anesthesiologists than in the general population.“

What is the evidence for this statement? What does "more severe" mean? Is the frequency per unit population higher, or is the severity of symptoms more severe?

2. What is the definition of “addiction” described in this paper? Is there a strict definition? Are the authors using it as a vague concept?

3. What is the difference between “addiction” and “abuse”? If they are exactly the same, why are the authors using two different words?

4.  p2, line 54, 55, “However, the abuse potential of propofol has been underestimated, and it has not been a controlled substance so far [16] “

The situation may differ from country to country.

5. p2, line 55-58, “It is reported that the number of propofol abuse increased five times from 1995 to 2005 [17]. Recently, propofol has been reported to become the most frequent abuse of anesthesia drugs among Australian and New Zealand anesthetists, accounting for 41% of drug abuse cases from 2004 to 2013 [10]. “

Reports from 10 years ago are not recent. Do the authors have more recent data? 

6. p5, line 201 “While the reward system is related to positive reinforcing, the anti-reward system is related to negative reinforcing in the central nervous system (CNS). The dysfunction of the reward system and the anti-reward system eventually leads to the formation and recurrence of addictive behaviors [93,94]. “

I don't quite understand the meaning of this sentence. Self-administration may occur as a result of the reward system functioning with propofol, as described by the authors “Like other addictive drugs, repeated propofol exposure can activate the reward system. p4, line153”. If the reward system becomes dysfunctional due to the antagonist of DAR1, self-administration may not occur.

7. p5, line196~  The authors state that the DAergic system in the NAc is involved in the maintenance of propofol self-administration and its rewarding effects, while the antagonist of DAR1 (SCH23390) and ERK (U0126) attenuated the self-administration of propofol [69,73,74]. The authors also described that direct injection of glucocorticoid receptor (GR) agonists into the NAc promoted propofol addiction in rats, and the self-administration of propofol could also be blocked by GR antagonists.

If you block the DAergic system or the CRF system, the self-administration of propofol is blocked in either way. Shouldn't these blocked states be called “dysfunction”? I'm not quite sure what states the authors refer to as "function" and "dysfunction" in the reward and anti-reward systems.

8. Ketamine also blocks NMDARs on GABAergic interneurons, causing the disinhibition of DAergic neurons and the activation of the reward system. Apart from the mechanism of addiction formation by the activation of the DAergic system, is there any evidence that the changes in the brain structure and neural connections due to ketamine use described in 3.2.3 contribute to the addiction formation?

9. p9, “The addictive drugs can be divided into three groups through their mechanism of increasing DA in the midbrain margin:(1) some drugs can reduce the inhibitory afferent of gamma GABAergic neurons to DAergic neurons, which is defined as disinhibition (opioids, γ-hydroxybutyric acid, and cannabinoids). (2) The second group can depolarize DAergic neurons directly by activating the α4β2 acetylcholine receptor (nicotine). (3) The third group is those drugs that can target DA transporters and interfere with DA reuptake (cocaine, amphetamines, and ecstasy) “

Does propofol fall into any of these three categories?

10. p12, line420,

“In addition, volatile anesthetics such as N2O can inhibit signal transduction in the neural network by enhancing the transmission of inhibitory signals (GABA, glycine) and reducing the transmission of excitatory neurotransmitters (nicotinic acetylcholine, NMDA, and opioid receptors) [165-168]. “

Is this sentence correct?  Are “opioid receptors” okay?

If N2O reduces the transmission of excitatory neurotransmitters (nicotinic acetylcholine), wouldn't addiction not occur?

Author Response

The authors describe non-opioid drug addiction in anesthesia care providers. Although it may be of little interest to the general public, it is a topic of great interest to health care providers exposed to these drugs. The authors seemingly have a good overview of the landscape of non-opioid anesthetic addiction. However, there seems to be a logical contradiction here and there. This paper generally based on statistics from more than 10 years ago. If the authors use “current” in the title, it should be supported by more recent (less than 5 years) data.

Response: We gratefully appreciate for the precious time the Reviewer spent making constructive remarks. We feel sorry for the inconvenience brought to the Reviewer. We have tried our best to update part of the references regarding the current situation of non-opioid drug addiction based on the Reviewer’s comments. However, there are no new large retrospective studies (less than 5 years) on the incidence of non-opioid anesthetic addiction among anesthesiologists. Therefore, we have retained part of these references.

Point 1: p1, line11 “Drug abuse is more severe in anesthesiologists than in the general population.” What is the evidence for this statement? What does "more severe" mean? Is the frequency per unit population higher, or is the severity of symptoms more severe?

Response 1: We appreciate the Reviewer’s professional comments and apologize for our confusing expression. To avoid misunderstanding, we have corrected the expression as follows: “Drug abuse is more common in anesthesiologists than in the general population because of their easier access to controlled substances.”

Point 2: What is the definition of “addiction” described in this paper? Is there a strict definition? Are the authors using it as a vague concept?

Response 2: Thank you for the above suggestions. In this review, we defined “addiction” as “Addiction is a chronically relapsing brain disease, which is related to the complex interactions between biological factors (pharmacological effects of drugs, genetics, ep-igenetics, developmental attributes, and neurocircuitry) and environmental factors (social and cultural systems, insufficient social connection, stress, trauma, and exposure to alternative reinforcers) [1,2]. The characteristics of addiction are compulsively seeking and taking drugs, loss of control when limiting intake, and negative emotions (such as irritability, anxiety, and dysphoria) when drugs are unavailable [3,4].” according to the definition of addiction proposed by American Society of Addiction Medicine in 2011.

Point 3: What is the difference between “addiction” and “abuse”? If they are exactly the same, why are the authors using two different words?

Response 3: We sincerely thank the Reviewer for the professional comments. The difference between “addiction” and “abuse” is very slight. Abuse is the excessive or improper use of a substance like alcohol or drug, which can lead to negative consequences, while addiction is a chronic disease characterized by compulsive drug seeking and use, despite harmful consequences. Although the conceptions of “abuse” and “addiction” are not exactly the same, they are often used interchangeably in many studies.

Point 4: p2, line 54, 55, “However, the abuse potential of propofol has been underestimated, and it has not been a controlled substance so far [16]” The situation may differ from country to country.

Response 4: We gratefully appreciate for your valuable suggestion. We have re-written this part according to the Reviewer’s comments in the revised manuscript as follows.

“However, the abuse potential of propofol has been underestimated, and it has not been a controlled substance anywhere in the world except in Korea so far [16,17]”

Point 5: p2, line 55-58, “It is reported that the number of propofol abuse increased five times from 1995 to 2005 [17]. Recently, propofol has been reported to become the most frequent abuse of anesthesia drugs among Australian and New Zealand anesthetists, accounting for 41% of drug abuse cases from 2004 to 2013 [10].” Reports from 10 years ago are not recent. Do the authors have more recent data? 

Response 5: Thank you for the above suggestion. Through our careful literature search again, we did not retrieve any more recent data regarding the incidence of propofol abuse. Thus, we have modified the corresponding description as follows: “It is reported that the number of propofol abuse increased five times from 1995 to 2005 [17]. In a retrospective study, propofol has been reported to become the most frequent abuse of anesthesia drugs among Australian and New Zealand anesthetists, accounting for 41% of drug abuse cases from 2004 to 2013 [10]”.

Point 6: p5, line 201 “While the reward system is related to positive reinforcing, the anti-reward system is related to negative reinforcing in the central nervous system (CNS). The dysfunction of the reward system and the anti-reward system eventually leads to the formation and recurrence of addictive behaviors [93,94].” I don't quite understand the meaning of this sentence. Self-administration may occur as a result of the reward system functioning with propofol, as described by the authors “Like other addictive drugs, repeated propofol exposure can activate the reward system. p4, line153”. If the reward system becomes dysfunctional due to the antagonist of DAR1, self-administration may not occur.

Response 6: We sincerely thank the Reviewer for your sincere review and professional suggestion. We are so sorry for our confusing expression. We have re-written this part according to the Reviewer’s comment. “While the reward system is related to positive reinforcing, the anti-reward system is related to negative reinforcing in the central nervous system (CNS). The changes of the reward system and the anti-reward system eventually lead to the formation and recurrence of addictive behaviors [93,94]”

Point 7: p5, line196~ The authors state that the DAergic system in the NAc is involved in the maintenance of propofol self-administration and its rewarding effects, while the antagonist of DAR1 (SCH23390) and ERK (U0126) attenuated the self-administration of propofol [69,73,74]. The authors also described that direct injection of glucocorticoid receptor (GR) agonists into the NAc promoted propofol addiction in rats, and the self-administration of propofol could also be blocked by GR antagonists. If you block the DAergic system or the CRF system, the self-administration of propofol is blocked in either way. Shouldn't these blocked states be called “dysfunction”? I'm not quite sure what states the authors refer to as "function" and "dysfunction" in the reward and anti-reward systems.

Response 7: We apologize for the incorrect expression and greatly appreciate the Reviewer’s comment. To avoid misunderstanding, we have corrected the description of “dysfunction” into “changes” in the revised manuscript.

Point 8: Ketamine also blocks NMDARs on GABAergic interneurons, causing the disinhibition of DAergic neurons and the activation of the reward system. Apart from the mechanism of addiction formation by the activation of the DAergic system, is there any evidence that the changes in the brain structure and neural connections due to ketamine use described in 3.2.3 contribute to the addiction formation?

Response 8: We greatly appreciate the Reviewer’s comment. The non-invasively technology to image the structure and function of the brain has deepened our understanding of the mechanism of addiction. Existing reviews of the extant literature have identified some of the major cortical and subcortical components of the complex neurobiological circuitry that forms the substrate for the compulsive seeking, excessive consumption, and negative emotions accompanying withdrawal that characterize addictive behaviors (Berman S, et al., Addict Rev, 2008; Goldstein R. Z., et al., Nat Rev Neurosci, 2011;). Moreover, previous studies do suggest that repeated ketamine use has the potential to alter brain structure and function (Strous JFM, et al., Front Neuroanat, 2022). Many studies have indicated that the changes in the brain structure and neural connections were associated with ketamine addiction. For example, prefrontal gray matter reductions may have been initiated by ketamine use, further impairing inhibition and facilitating ketamine dependence. However, non-invasive neuroimaging is only one part of a wider effort to understand the etiology of addiction. Still, the question remains of how these changes in the brain structure and neural connections modulate the process of addiction, and further studies are still needed on the mechanisms underlying the impact of these changes on addiction.

Point 9: p9, “The addictive drugs can be divided into three groups through their mechanism of increasing DA in the midbrain margin:(1) some drugs can reduce the inhibitory afferent of gamma GABAergic neurons to DAergic neurons, which is defined as disinhibition (opioids, γ-hydroxybutyric acid, and cannabinoids). (2) The second group can depolarize DAergic neurons directly by activating the α4β2 acetylcholine receptor (nicotine). (3) The third group is those drugs that can target DA transporters and interfere with DA reuptake (cocaine, amphetamines, and ecstasy)” Does propofol fall into any of these three categories?

Response 9: We sincerely thank the reviewer for the professional comments. Several studies have demonstrated that propofol addiction can significantly increase the DA levels in NAc. However, these studies could not differentiate among disinhibition, increased release of DA or inhibition of DA re-uptake that resulted in increased DA levels. Thus, based on current limited research, we may not be able to classify propofol into any of these three categories.

Point 10: p12, line420, “In addition, volatile anesthetics such as N2O can inhibit signal transduction in the neural network by enhancing the transmission of inhibitory signals (GABA, glycine) and reducing the transmission of excitatory neurotransmitters (nicotinic acetylcholine, NMDA, and opioid receptors) [165-168]. ”Is this sentence correct? Are “opioid receptors” okay? If N2O reduces the transmission of excitatory neurotransmitters (nicotinic acetylcholine), wouldn't addiction not occur?

Response 10: We appreciate the Reviewer’s comments and apologize for our incorrect description. Volatile anesthetics can inhibit signal transduction in the neural network by enhancing the transmission of inhibitory signals (GABA, glycine) and reducing the transmission of excitatory neuro-transmitters (nicotinic acetylcholine, NMDA, and opioid receptors). However, it is uncertain whether these effects are related to addiction. Therefore, to avoid misunderstandings, we have deleted the relevant description in the revised manuscript according to the Reviewer’s comments.

Reviewer 2 Report

This is an important and will be a nice contribution to the journal and the scientific community as a whole

Author Response

This is an important and will be a nice contribution to the journal and the scientific community as a whole

Response: We thank the reviewer for reading our paper carefully and giving the above positive comments.

Reviewer 3 Report

Dear Authors:

Your manuscript entitled Non-opioid anesthetics addiction: a review of current situation and mechanism is not well structured

Comments and suggestions:

Abstract

I suggest that the abstract there should more clearly stated the main aims, possible novelties and/or contributions and implications of the study.

1. Introduction

This section is without clear motivation, without a focus and without research questions. It is suggested to specify in a better way the motivation, aims and objectives, and possible novelty and/or contribution of the manuscript to the literature.

2. Literature Review and Theoretical Framework

This section is absent, I suggest to report it and with the aim to extend the literature review

I suggest to report the Research Questions (RQs) in the section.

Introduction should be brief, providing motivation of the research and outline main research focus. The objectives must be specified more in detail.

3. Materials and Methods

This section is absent, I suggest to report it

Conclusion

Conclusions and recommendations should be improved as they largely repeated the results. The character of conclusion is too general one and it repeats results. Authors should better underline conclusions, and intentions for future researches should be noted at the end of the conclusions.

What are the proposals for research in future?

Author Response

Your manuscript entitled Non-opioid anesthetics addiction: a review of current situation and mechanism is not well structured

Response: We thank the Reviewer for pointing out this issue. We have modified the structure according to the Reviewer’s suggestion in the revised manuscript.

Point 1: Abstract: I suggest that the abstract there should more clearly stated the main aims, possible novelties and/or contributions and implications of the study.

Response 1: We sincerely appreciate the valuable comments. We have re-written the abstract based on the Reviewer’s suggestion in the revised manuscript.

Point 2: Introduction: This section is without clear motivation, without a focus and without research questions. It is suggested to specify in a better way the motivation, aims and objectives, and possible novelty and/or contribution of the manuscript to the literature.

Response 2: We sincerely thank the Reviewer for the professional comments. We have re-written the introduction in a better way the motivation, aims and objectives, and possible novelty and/or contribution of the manuscript to the literature according to the Reviewer’s comment.

Point 3: Literature Review and Theoretical Framework: This section is absent, I suggest to report it and with the aim to extend the literature review. I suggest to report the Research Questions (RQs) in the section. Introduction should be brief, providing motivation of the research and outline main research focus. The objectives must be specified more in detail.

Response 3: Thanks for your careful review work and professional suggestions. We have modified the Introduction based on the Review’s comments and requirements of the journal in the revised manuscript, which clearly describes the aim, research questions, research motivation, and the main research focus.

Point 4: Materials and Methods: This section is absent, I suggest to report it

Response 4: We sincerely appreciate the valuable comments. We have added the section of Methods in the revised manuscript as follows.

The manuscript includes research articles gathered from PubMed that expounded the clinical situation and mechanism of the Non-opioid anesthetics addiction. The following key words were used: “anesthesiology”, “drug addiction”, “substance use disorders”, “propofol addiction”, “ketamine addiction”, “benzodiazepines addiction”, and “inhalational anesthetics addiction”. Only articles written in English were selected for study.

Point 5: Conclusion: Conclusions and recommendations should be improved as they largely repeated the results. The character of conclusion is too general one and it repeats results. Authors should better underline conclusions, and intentions for future researches should be noted at the end of the conclusions. What are the proposals for research in future?

Response 5: Thanks a lot for your professional advice. We have modified the Conclusions according to the Reviewer’s comments in the revised manuscript as follows.

Increasing evidence indicated that the incidence of abuse of non-opioid anesthetic drugs (including propofol, ketamine, BDZs, and inhalational anesthetics) is increasing, which needs more attention from researchers and exposed patients. Effective regulatory and therapeutic methods are needed for the identification, monitoring, and treatment of individuals with non-opioid anesthetic drug abuse. Understanding the current situation and mechanism of non-opioid anesthetics addiction is helpful to improve a better understanding of it and can provide certain ideas for the prevention and treatment of non-opioid anesthetics. Growing studies have confirmed the abuse potential for propofol, ketamine, BDZs, and inhalational anesthetics using CPP and SA models. Like other classic addictive, propofol, ketamine, BDZs, and inhalational anesthetics could also influence the reward system. However, the focus on the clinical situation and the potential mechanisms of non-opioid anesthetics addiction appears to be in its preliminary stages.

In the future, more clinical studies are needed to update the incidence of non-opioid anesthetics addiction and determine the population at risk of non-opioid anesthetics addiction abuse. Besides, more advanced imaging techniques, tracing techniques, optogenetic tools, chemogenetic tools, and electrophysiological techniques should also be used to analyze the complex neural network behind the specific reward circuitry of non-opioid anesthetics to further clarify the specific mechanism of the non-opioid anesthetics addiction.

Reviewer 4 Report

The paper present a bibliographic revision about an interesting topic, do not present any noticeable scientific novelty but present a clear and well structured revision, however some aspects need to be corrected, the author present as recent data publications from 2013 and 2015 (references 5 and 8), these are not recent years. the figures are clear.

Author Response

Point : The paper present a bibliographic revision about an interesting topic, do not present any noticeable scientific novelty but present a clear and well structured revision, however some aspects need to be corrected, the author present as recent data publications from 2013 and 2015 (references 5 and 8), these are not recent years. the figures are clear.

Response: We gratefully thank you for the precious time the reviewer spent making constructive remarks. We have replaced the old literature (previous references 5 and 8) with the recent literature in the revised manuscript.

Round 2

Reviewer 1 Report

The manuscript was significantly improved.